# Vesicoamniotic Shunting before 17 + 0 Weeks in Fetuses with Lower Urinary Tract Obstruction (LUTO): Comparison of Somatex vs. Harrison Shunt Systems

**DOI:** 10.3390/jcm11092359

**Published:** 2022-04-22

**Authors:** Brigitte Strizek, Theresa Spicher, Ingo Gottschalk, Paul Böckenhoff, Corinna Simonini, Christoph Berg, Ulrich Gembruch, Annegret Geipel

**Affiliations:** 1Department of Obstetrics and Prenatal Medicine, University Hospital Bonn, Venusberg Campus 1, 53127 Bonn, Germany; brigitte.strizek@ukbonn.de (B.S.); theresaspicher@web.de (T.S.); paul.boeckenhoff@ukbonn.de (P.B.); corinna.simonini@ukbonn.de (C.S.); ulrich.gembruch@ukbonn.de (U.G.); 2Division of Prenatal Medicine, Department of Obstetrics and Gynecology, University Hospital Cologne, 50937 Cologne, Germany; ingo.gottschalk@uk-koeln.de (I.G.); christoph.berg@ukbonn.de (C.B.)

**Keywords:** LUTO, megacystis, vesicoamniotic shunting, posterior urethal valve, fetal therapy

## Abstract

(1) Background: The aim of this study was to compare perinatal outcomes and complication rates of vesicoamniotic shunting (VAS) before 17 + 0 weeks in isolated LUTO (lower urinary tract obstruction) with the Somatex^®^ intrauterine shunt vs. the Harrison fetal bladder shunt. (2) Methods: This is a retrospective cohort study in two tertiary fetal medicine centers. From 2004–2014, the Harrison fetal bladder shunt was used, and from late 2014–2017, the Somatex shunt. Obstetrics and pediatric charts were reviewed for complications, course of pregnancy, perinatal outcome, and postnatal renal function. (3) Results: Twenty-four fetuses underwent VAS with a Harrison (H) shunt and 33 fetuses with a Somatex (S) shunt. Live birth rates and survival to last follow-up were significantly higher in the Somatex group, at 84.8% and 81.8%, respectively, vs. 50% and 33.3% in the Harrison group (*p* = 0.007 and *p* < 0.001). The dislocation rate in the Somatex group (36.4%) was significantly lower than in the Harrison group (87.5%) (*p* < 0.001). The median time to dislocation was significantly different, at 20.6 days (H) vs. 73.9 days (S) (*p* = 0.002), as was gestational age at dislocation (17 (H) vs. 25 (S) weeks, *p* < 0.001). Renal function was normal in early childhood in 51% (S) vs. 29% (H) (*p* = 0.11). (4) Conclusions: VAS before 17 + 0 weeks gestational age with a Somatex shunt improves perinatal survival significantly and might even have a positive effect on renal function, probably due to the lower dislocation rates. A normal amount of amniotic fluid in the third trimester was the best predictor of normal renal function in early childhood.

## 1. Introduction

Lower urinary tract obstruction is rare after birth (2–3/10,000), but megacystis has been reported in 1 of 1345 in a first trimester cohort [1]. The most common etiologies of lower urinary tract obstruction (LUTO) are posterior urethral valves (PUV), but there are also other obstructive (e.g., urethral stenosis or atresia, cloacal malformations) and non-obstructive causes (e.g., megacystis-microcolon-intestinal hypoperistalsis syndrome (MMIH), neurological etiologies) [1,2,3].

Even after the exclusion of aneuploidy and severe additional malformations, counseling is difficult when faced with large fetal megacystis in the first or early second trimester and the prognosis is often conveyed to be limited, shown by the high number of terminations of pregnancy (TOP) in first trimester cohorts [2,3,4]. Among other reasons, this is due to the lack of early therapeutic options and uncertainty of candidate selection.

Vesicoamniotic shunting (VAS) in the second trimester has been shown to increase neonatal survival due to attenuation of pulmonary hypoplasia, but the rates of long-term renal impairment remain high [5,6,7]. Animal data show that glomerulo- and nephrogenesis are impaired after obstruction of the lower urinary tract and earlier onset and longer duration of the obstruction significantly worsen renal dysplasia; this is one reason why VAS in the late second trimester is not likely to show a positive effect on renal function. In humans, nephrogenesis is 80% complete by 22–24 weeks gestational age (GA) and fully complete by 34–36 weeks GA.

In our opinion, any intervention aiming to reduce renal impairment should be performed as early as possible. There is considerable controversy over VAS before 16–17 weeks GA, due to high complication and shunt dislocation rates, but very few studies have so far evaluated VAS in the early second trimester [8,9,10].

The aim of this study was to compare fetal and maternal outcomes of vesicoamniotic shunting before 17 weeks using two different shunts: the Harrison fetal bladder stent and the Somatex intrauterine shunt.

## 2. Materials and Methods

We conducted a retrospective cohort study in two tertiary fetal medicine centers from 2004–2017. We included male fetuses with a diagnosis of isolated megacystis on initial prenatal ultrasound and VAS not later than 16 + 6 weeks of gestation. VAS was offered from 14 + 0 weeks before 2014 and from 12 + 0 weeks thereafter. The time period was divided according to the type of shunt that was used.

Megacystis was defined as a longitudinal bladder diameter of >15 mm in the mid-sagittal plane in the first trimester and >25 mm in the second trimester. 

All fetuses received systematic first and/or second trimester sonographic evaluation, which included nuchal translucency measurement in the first trimester and echocardiography. Vesicocentesis was not performed before VAS and urine biochemistry was not a prerequisite prior to VAS. All fetuses included in the study had a normal karyotype, no additional anomalies on ultrasound, and no evidence of renal dysplasia at the time of the intervention. Parents received multidisciplinary counseling by fetal medicine specialists, neonatologist, and pediatric nephrologists, including on the uncertainty of the effect of VAS on renal function. Written informed consent was obtained from all patients before the procedure. 

Patients were followed regularly by ultrasound for shunt dislocation and VAS was repeated as soon as possible after dislocation occurred.

Shunt insertion was performed percutaneously under ultrasound guidance by experienced operators with more than 5 years of experience in fetal intervention. 

The Harrison fetal bladder stent (Cook Medical Inc., Bloomington, IN, USA) is a double pigtail stent with an outer diameter of 5 French (1.67 mm) and inner diameter of 0.97 mm. The usable length between the pigtails is 15–35 mm; it is introduced through a 13 G needle. Local anesthesia was used in patients receiving a Harrison shunt (Figure 1).

The Somatex^®^ intrauterine shunt (Somatex Medical Technologies GmbH, Berlin, Germany) is 25 mm long with a diameter of 2.6 mm for the expanded shunt, consisting of a nitinol wire mesh and internal impermeable silicone coating. The shunt has self-deploying parasols at both ends and can be placed through an 18 G puncture cannula (Figure 1). The Somatex intrauterine shunt was available in Germany from late 2014.

Obstetric and pediatric electronic databases and charts were reviewed for associated anomalies, prenatal course of pregnancy, amount of amniotic fluid during pregnancy, prenatal and neonatal complications, neonatal outcome, and interventions after birth. Neonatal survival was defined as being alive at 28 days. Renal function was assessed by local age-dependent reference values for creatinine and urea. End-stage renal failure was defined as the need for dialysis or transplantation.

Patients that underwent VAS before 2012 (*n* = 18) were included in a previous publication [9] and the outcome of VAS before 14 weeks with a Somatex shunt has been published as a feasibility study (*n* = 10) [11].

Statistical analysis was performed using Student’s *t*-test and Fisher’s exact test; a *p*-value of <0.05 was considered statistically significant. The study was approved by the ethics committees of the universities of Bonn (#127/18) and Cologne (#19-1186).

## 3. Results

In the entire study period, 57 patients underwent VAS before 17 weeks GA: 24 patients with a Harrison fetal bladder shunt (2004–2014) and 33 patients with a Somatex^®^ intrauterine shunt (from late 2014–2017) (Figure 2).

### 3.1. Harrison Fetal Bladder Stent (n = 24) 

Shunt insertion was performed at a median gestational age of 14 + 4 weeks (13 + 1 to 16 + 6 weeks) (Table 1). In 13 fetuses, a keyhole sign was present. The amount of amniotic fluid prior to VAS was considered normal in 10 cases (41%). Amnioinfusion was necessary in 13 (54.2%) patients to facilitate VAS; this was necessary in 3 patients despite normal amniotic fluid.

There were 4 (16.7%) miscarriages or intrauterine fetal deaths (IUFD) and 8 (33.3%) terminations of pregnancy (TOP). Reasons for TOP were shunt related in 7/8 patients, including dislocation, anhydramnios due to suspected renal failure, and PPROM. In one patient, TOP was due to additional malformations that were detected later on in pregnancy. Oligohydramnios in the third trimester (>27 weeks) developed in 5 of the continuing pregnancies (5/12, 41.7%). 

#### 3.1.1. Shunt Complications 

Shunt dislocation occurred in 21 patients (87.5%) at a median interval of 20.6 days (range 1–111 days) after initial VAS and a median gestational age of 17 + 6 weeks (14 + 1−30 + 2 weeks). Dislocation occurred in 19 fetuses before 24 weeks, representing 79.2% (19/24) of the entire group. A re-intervention was performed in 8/24 patients (33.3%). The mean number of shunts per fetus was 1.6, with a maximum of 4 shunts in 1 patient. Preterm premature rupture of membranes (PPROM) occurred in 6/24 patients (25%). Other than two cases of chorioamnionitis after PPROM, no maternal complications were recorded.

#### 3.1.2. Neonatal Outcome 

Twelve (50%) children were born alive (Table 2). There were 3 neonatal deaths (NND) due to pulmonary hypoplasia: 1 case was in an infant born prematurely at 27 weeks; in another case, the parents opted for palliative care due to lung hypoplasia and suspected renal failure.

Median GA at birth of the 11 neonates with intention of postnatal treatment was 37 + 0 weeks (27 + 0 to 40 + 4 weeks). Five children were born before 37 weeks GA.

Adverse outcome (TOP, miscarriage, IUFT, NND) occurred in 12 patients (50%). Overall perinatal survival was 37.5% (9/24) and 56% after exclusion of TOP (9/16). 

There was one childhood death due to severe pulmonary insufficiency after H1N1 influenza during the second year of life.

The postnatal diagnosis was posterior urethral valve (PUV) in four of the boys. Urethral stenosis and vesico-urethral reflux (VUR) were found in two boys and prune belly syndrome in one boy. In another child, so far there was no evidence of any urological pathology. Additional postnatal diagnoses in survivors included Smith–Lemli–Opitz syndrome in one and coarctation of the aorta in another (Table 3).

### 3.2. Somatex^®^ Intrauterine Shunt (n = 33) 

Shunt placement was performed at a median gestational age of 15 + 1 weeks (12 + 4 to 16 + 6 weeks), which was not statistically different from the Harrison shunt group (Table 1). In 25 fetuses, there was a keyhole sign. There was normal amniotic fluid volume prior to VAS in 18 cases (54.5%). Amnioinfusion prior to VAS was performed in 12 (36.4%) cases. There were 4 TOP (12.2%) and 1 IUFD at 17 + 4 weeks (3%). TOP was performed either due to additional malformations that were detected later in pregnancy (*n* = 2), shunt complications (early PPROM, *n* = 1), or signs of severe renal dysplasia (*n* = 1) during pregnancy. Oligohydramnios >27 weeks was recorded in 11 patients (39.3%, 11/28). 

#### 3.2.1. Shunt Complications 

Shunt dislocation occurred in 12 of 33 patients (36.4%) at a median GA of 25 + 6 weeks (15 + 2−37 + 4 weeks). The median time interval to dislocation was 73.9 days (1–160 days) after first VAS. Dislocation occurred in 3 cases <24 weeks, which represented 9.1% (3/33) of the entire group. A re-intervention was performed in 9/33 (27.3%) pregnancies, 6 received an abdominoamniotic shunt due to urinary ascites. The mean number of shunts per fetus was 1.3, with a maximum of three shunts. 

The shunt dislocated externally in the abdominal wall in one fetus and into the fetal bladder in another. Nine shunts dislocated into the abdominal cavity of the fetus. In one patient, the external end of the shunt remained stuck in the uterine wall. The shunt migrated into the maternal abdomen during pregnancy without causing further complications, and was removed by laparotomy after an uneventful vaginal delivery. There were no other maternal complications. PPROM at any time during pregnancy occurred in 7/33 (21.2%) cases. 

In 11 neonates, postnatal surgery to remove the shunt was necessary. One child showed omental prolapse through an iatrogenic abdominal wall defect after birth; complications after the operation ultimately led to short bowel syndrome.

#### 3.2.2. Neonatal Outcome 

Twenty-eight neonates were born alive (84.8%) at a median gestational age of 36 + 0 weeks (31 + 0−41 + 0 weeks) (Table 2). There were 15 (53.6%) premature deliveries before 37 + 0 weeks. There was one NND (3.6%) due to pulmonary hypoplasia. Adverse outcome (TOP, IUFD, NND) was recorded in six pregnancies (18.2%). Perinatal survival was 81.8% (27/33) and 93.1% (27/29) after the exclusion of TOP. 

PUV was the final diagnosis in 48.1% (13/27) of survivors and urethral stenosis/atresia was found in 11 (40.7%). Two children (7.4%) were diagnosed with megacystis-microcolon-intestinal hypoperistalsis syndrome (MMIHS) after birth and one child had no urological pathology. There were signs of renal dysplasia on ultrasound in at least one kidney in 9 of 27 children. One child had a unilateral nephrectomy after recurrent infections in the non-functioning kidney, with good function of the contralateral kidney.

Four children (14.8%) had anorectal malformations, one of which had a very severe form (cloacal dysgenesis). Other associated malformations are shown in Table 3. One child had cerebral palsy and severe developmental delay. 

### 3.3. Renal Function 

Follow-up regarding renal function for >1 year was available for 9 children in the Harrison group and for 25 children in the Somatex group. Last follow-up was at a median age of 7.3 years (1.1–11.2 years) in the Harrison group and 2.9 years (1.4–4.7 years) in the Somatex group. Two children in the Somatex group were lost to follow-up and one child in the Harrison group died during the second year of life; therefore, data were available for 8 and 25 children, respectively, at the last follow-up.

At the last follow-up, 87.5% (7/8) of children in the Harrison group and 68% (17/25) in the Somatex group had normal renal function, which is not a statistically significant difference (*p* = 0.39). There was a greater difference in rates of normal renal function on an intention-to-treat basis, but this also did not reach statistical significance: Harrison 29% (7/24) vs. Somatex 51.5% (17/33) (*p* = 0.11).

Two children in the Harrison group (22.2%) required dialysis. One of those infants died during childhood, the other received a kidney transplant from a relative. 

In the Somatex group, two children had mild renal impairment and six children had severe renal insufficiency or end-stage renal failure. Two had already undergone transplantation, two were on dialysis, and two were not receiving renal replacement therapy. 

Normal renal function was statistically more common if amniotic fluid was normal after 27 weeks in the entire population: 3/16 (18.8%) vs. 21/22 (95.5%) (*p* < 0.001), irrespective of the type of shunt.

## 4. Discussion

Vesicoamniotic shunting is the most common intervention in fetuses with LUTO in the literature, but there are very few reported cases of VAS before 17 weeks, as it has been associated with very high rates of complications. Here, we present the largest cohort study of vesicoamniotic shunting for LUTO before 17 weeks of gestation. In addition, we compared outcomes according to the type of shunt used: Harrison fetal bladder stent vs. Somatex^®^ intrauterine shunt. 

The prognosis of untreated megacystis diagnosed before 17 weeks in the literature is catastrophic, even after the exclusion of aneuploidies and additional malformations. The majority of pregnancies are terminated and in the remaining cases there is a high probability of perinatal death due to pulmonary hypoplasia [2,3]. Early diagnosis in itself seems to be associated with a worse prognosis, as more severe forms of obstruction as urethral stenosis or atresia appear to be more prevalent. Even in the most recent studies of the natural history with a diagnosis before 17 weeks, the live birth rate of severe LUTO remains low, at around 36% [12]. Considering these data, counseling parents after detection of a severe form of LUTO in the first trimester remains difficult and the uncertainty regarding long-term outcome and lack of technically feasible, effective, and safe interventions contributes to more than half of the patients deciding to terminate the pregnancy. 

VAS in the late second trimester has been shown to improve survival, mainly by lowering rates of lethal pulmonary hypoplasia, without having a proven protective effect on renal function [5,7,13,14]. Animal studies have shown that the time of onset and the duration of the obstruction are critical for the development of renal dysplasia. Signs of renal dysplasia have been found in post-mortem examinations as early as 15 weeks GA [4]. Therefore, VAS in the late second trimester might come too late to salvage renal function. Faced with the dilemma of expectant management until 17 weeks and thereby risking worsening renal function versus potentially higher complication rates, we have adopted a policy of performing VAS as soon as possible after the diagnosis and appropriate counseling. When the Somatex shunt became available in 2014, we decided to use this type of shunt due to its smaller introduction cannula of 18 G and its assumed design advantages (better visualization on ultrasound, self-deploying parasols to prevent dislocation). 

In our cohort, the dislocation rate of the Somatex shunt (36.4%) was significantly lower compared to the Harrison shunt (87.5%) (*p* < 0.001). In addition, the median time to dislocation was also vastly different: 21 days (Harrison) vs. 74 days (Somatex) (*p* = 0.002) as was gestational age at dislocation (17 vs. 25 weeks, *p* < 0.001). The rate of dislocation before 24 weeks was significantly lower in the Somatex group (9%) vs. the Harrison group (79%) (*p* < 0.001). The rate of re-interventions did not differ in both groups with 33% (Harrison) vs. 27% (Somatex). However, repeated shunt dislocation occurred more frequently after re-shunting with the Harrison shunt compared with the Somatex shunt (63% vs. 27%, *p* = 0.014).

Shunt related complications of the Somatex shunt in our cohort are comparable to what has been reported in the literature of VAS at later gestational ages. The dislocation rate was 20% in the PLUTO trial, with a 40% overall complication rate; others have reported dislocation rates of 52.6% [5,15]. The dislocation rates of the Harrison shunt are even higher in our experience: 87% in this study before 17 weeks and 66% in a previous publication [9]. Kurtz et al. have reported an equally high dislocation rate of 78% (7/9) for this type of shunt [15]. The time interval to dislocation of the Harrison shunt is also similarly short in other studies [15,16]. There is one other type of shunt, a double-basket shunt from Japan, that has an outer diameter of 1.47 mm. Jeong et al. reported their experience of 32 fetuses with a shunt dislocation or occlusion occurring in 18 of 42 procedures (42.8%) [17].

Live birth rates were significantly higher in the Somatex group (84.8%) versus 50% in the Harrison group (*p* = 0.007), as was survival at last follow-up (81.8% vs. 33.3%, respectively) (*p* < 0.001). VAS with a Somatex shunt before 17 weeks therefore seems to lead to even higher survival compared to survival rates (50–65%) after late second trimester VAS [5,6,7,13,14,18].

In the group treated with a Harrison shunt there were two neonatal deaths due to lung hypoplasia and one death due to lung hypoplasia and prematurity (3/12, 25%). In the Somatex group, the rate of neonatal death to due lung hypoplasia was significantly lower with 3.6% (1 of 28 live-born children) or 6% (2/33) including TOP due to renal dysplasia and suspected lung hypoplasia. We speculate that this difference is due to a higher proportion of pregnancies with a sufficient amount of amniotic fluid during the critical time of lung development in the Somatex group, because of the lower overall rate of shunt dislocations and longer time interval to dislocation. Even in survivors that were later found to have renal dysplasia, oligo/anhydramnios rarely occurred before 24 weeks, which might explain the low rate of lethal lung hypoplasia. 

Oligo/anhydramnios at >27 weeks of gestation was the best predictor of renal impairment after birth. In pregnancies with normal amniotic fluid beyond 27 weeks, there was no neonate with severe renal insufficiency requiring dialysis or renal transplantation in both groups. If there was oligohydramnios in the third trimester, the rates of adverse outcome (NND, renal failure, TOP) were, however, considerably higher.

Compared to the results of our feasibility study of VAS with a Somatex shunt at <14 weeks GA [11], perinatal survival seems to be similar to the results of the entire cohort before 17 weeks, with 75% after the exclusion of TOP and one NND due to lung hypoplasia. Dislocation rates were also similar for VAS at <14 weeks GA (40%, 4/10) and VAS between 14 + 0−16 + 6 weeks (34.8%, 8/23). Our results appear similar to a study of the Japanese double-basket catheter, which reported an overall perinatal survival rate of 68.8% (22/32) and normal renal function in 40% of the children (10/25) at 2 years after birth [17].

We did not perform fetal urine analysis prior to VAS as there are no normalized data for first trimester fetal urine. In addition, due to the design of the Somatex shunt, sampling during VAS is not possible and we chose to refrain from an additional vesicocentesis to reduce complication rates. The propositions of prenatal staging prior to VAS to exclude candidates with poor renal prognosis are mainly based on data from the late second trimester [19]. Signs of renal dysplasia are more subtle on ultrasound in the first trimester and therefore the prognostic value of the classification systems is even less certain in the populations with an early manifestation of LUTO. The use of staging classifications for candidate selection has been questioned by the recently published consensus of the ERKNet CAKUT-Obstructive Uropathy Work Group [20].

The main limitations of our study are the retrospective design and the different durations of follow-up in the two groups. However, death during follow-up was a rare event and only one child died during childhood in the Harrison group. When comparing the prevalence of renal impairment during the first year of life and beyond, there was only one child that developed mild renal insufficiency at three years of age in our cohort. In all other children, the classification of renal function did not change; therefore, we decided to compare both groups with regard to renal function, even if the length of follow-up was considerably different. However, long-term outcomes beyond early childhood are still missing for both groups and co-morbidities due to additional anorectal malformations should be considered. 

The long duration of the study is another limitation of our study. We cannot exclude that changing attitudes and practices, prenatally and in neonatal management, might have influenced decision-making, especially regarding termination of pregnancy in the earlier study period, as more parents opted for TOP if the first Harrison shunt dislocated within a short time. In addition, there was no standardized follow-up of live-born children, and they were treated at different pediatric nephrology centers. From the pediatric literature, we know that renal function at birth or the first months of life does not exclude worsening of renal function over the years, although we did not observe this during the short follow-up in our cohort. Although this is one of the largest studies of VAS in general, the sample size remains modest, and the results cannot be generalized. 

All the operators that performed VAS in our centers already had a large experience in the beginning of the study period, but individual operator experience might still have improved over time, which might have contributed to the lower dislocation rates in the more recent Somatex group, irrespective of the effect of the different type of shunt. 

In the past, dislocation and complication rates have never been compared by the type of shunts used, so the effect of the type of shunt itself remains largely unknown. Due to the different designs and materials, the type of shunt used might influence the outcome more than we have thought in the past. In the future, not only gestational age at VAS, but also the type of shunt and intervention, as well as the complication-free interval should be studied in addition to other potentially predictive factors, such as timing of oligohydramnios and signs of renal dysplasia.

Vesicoamniotic shunting before 17 weeks GA with the Somatex shunt leads to significantly higher rates of perinatal survival compared to the Harrison shunt. In our opinion, this is attributable to lower rates of shunt-related complications (miscarriage, PPROM, dislocation) and longer intervals to dislocation. The overall perinatal prognosis of male fetuses with isolated LUTO before 17 weeks is improved considerably by using a Somatex shunt for VAS compared to our historical control, but our results regarding renal function still must be regarded with caution. The rates of renal failure remain high in both groups when only survivors after birth are evaluated. On the other hand, in an intention-to-treat comparison, normal renal function was higher in the Somatex group (51% vs. 29%) but did not reach statistical significance.

## 5. Conclusions

In conclusion, VAS before 17 + 0 weeks GA with a Somatex^®^ intrauterine shunt was superior to the Harrison shunt in terms of perinatal survival and dislocation rates, and might even exert a positive effect on renal function. A normal amount of amniotic fluid in the third trimester was the best predictor of normal renal function in early childhood. With complication rates similar to VAS in the second trimester, we feel that VAS should be offered from the late first trimester in settings where the Somatex shunt is available. However, parents should be counseled by a multidisciplinary team about the uncertainty regarding long-term renal function and additional malformations.

## Figures and Tables

**Figure 1 jcm-11-02359-f001:**
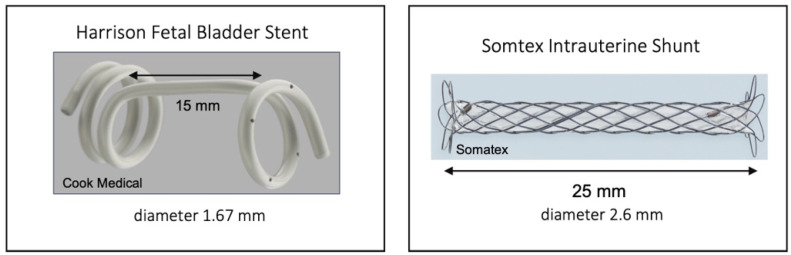
Types of shunts used for vesicoamniotic shunting.

**Figure 2 jcm-11-02359-f002:**
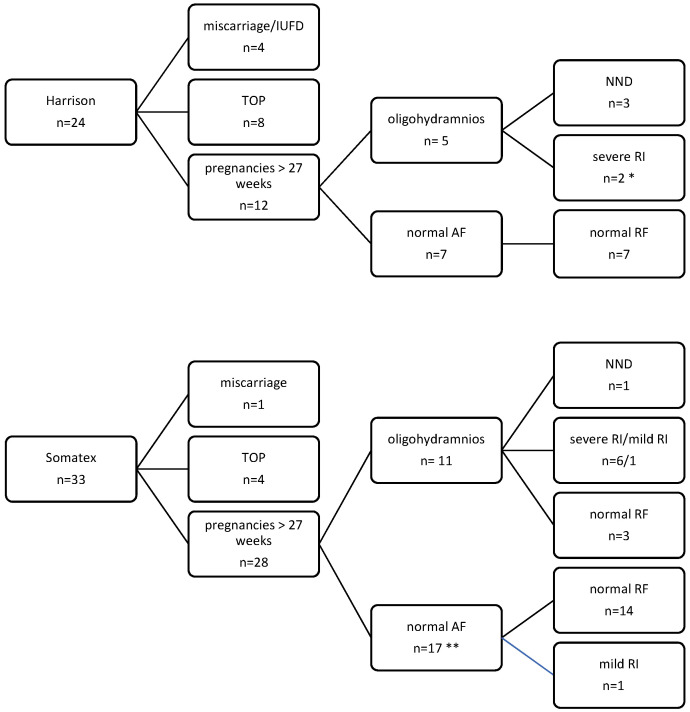
Outcome of pregnancies according to type of shunt. * 1 died during follow-up; ** 2 lost to follow up. AF (amniotic fluid); IUFD (intrauterine fetal death); NND (neonatal death); RI (renal impairment); RF (renal function); TOP (termination of pregnancy).

**Table 1 jcm-11-02359-t001:** Patient characteristics and shunt-related complications by type of shunt.

	Harrison (*n* = 24)		Somatex (*n* = 33)		*p*-Value
Median GA at first VAS (weeks, range)	14 + 4	(13 + 1−16 + 6)	15 + 1	(12 + 4−16 + 6)	0.496
Patients with shunt dislocation *n*, (%)	21	(87.5%)	12	(36.4%)	<0.001
Median GA at 1st dislocation (weeks, range)	17 + 6	(14 + 1−30 + 2)	25 + 6	(15 + 2−37 + 4)	<0.001
Median interval to first dislocation (days, range)	20.6	(1–111)	73.9	(1–160)	0.002
Patients with re-intervention *n*, (%)	8	(33.3%)	9	(27.3%)	0.771
Dislocations/total no. of shunts *n*, (%)	24/38	(63.2%)	12/44	(27.3%)	0.002
Complications * (excl. dislocation) *n*, (%)	13	(54.2%)	16	(48.5%)	0.790

GA (gestational age); NND (neonatal death); VAS (vesicoamniotic shunt); TOP (termination of pregnancy). * complications: PPROM (preterm premature rupture of membranes), chorioamniotic membrane separation, fetal bladder rupture, iatrogenic abdominal wall defect, miscarriage, IUFD (intrauterine fetal death). A *p*-value of <0.05 is considered significant.

**Table 2 jcm-11-02359-t002:** Outcome by type of shunt.

	Harrison	Somatex	
TOP/miscarriage/IUFD *n*, (%)	12/24 (50%)	5/33 (15.2%)	0.008
Live birth *n*, (%)	12/24 (50%)	28/33 (84.8%)	0.007
Neonatal death *n*, (%)	3/24 (12.5%)	1/33 (3%)	0.3
No. of survivors at last follow-up *n*, (%)	8/24 (33.3%)	27/33 (81.8%)	<0.001
No. of survivors with good renal function *n*, (%)	7/8 (87.5%)	17/25 (68%)	0.39
Good renal function/entire group *n*, (%)	7/24 (29.2%)	17/33 (51.5%)	0.11
Oligohydramnios > 27 weeks and good renal function *n*, (%)	0/5 (0%)	3/11 (27.3%)	0.51

TOP (termination of pregnancy); IUFD (intrauterine fetal death). A *p*-value of <0.05 is considered significant.

**Table 3 jcm-11-02359-t003:** Additional malformations in 36 survivors.

	Harrison (*n* = 9)	Somatex (*n* = 27)
Club foot/feet	1 (11.1%)	5 (18.5%)
VACTERL/caudal regression	1 (11.1%)	3 (11.1%)
Prune belly syndrome	2 (22.2%)	4 (14.8%)
Complex anorectal malformation Isolated anal atresia	2 (22.2%) 0	3 (11.1%) 1 (3.7%)
Smith–Lemli–Opitz syndrome	1 (11.1%)	0
Major cardiac anomalies	1 (11.1%) coarctation of the aorta	1 (3.7%) aorto-pulmonary window

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
