# Peer review of "Vesicoamniotic Shunting before 17 + 0 Weeks in Fetuses with Lower Urinary Tract Obstruction (LUTO): Comparison of Somatex vs. Harrison Shunt Systems"

_jcm, 2022, doi:10.3390/jcm11092359_

Round 1

Reviewer 1 Report

This is a retrospective study on 57 fetuses with isolated megazystis on prenatal ultrasound that had vesicoamniotic shunt placement before 17 weeks gestation. The authors compare a historic cohort with the Harrison shunt system to their present Somatex system used after 2014 and report on complication rates as well as perinatal outcome and long term outcome. They show that the Somatex system has lower rates of dislocation and a higher perinatal survival rates. No significant differences were found in regard to shunt complications and percentage of survivors with good renal function. The study has the known limitations in regard to a retrospective study design and comparison to a historic cohort which is mentioned in sufficient detail in the discussion.

The study is well written, the methods and results are clear and the discussion is interesting, includes the important other studies, discusses the limitations in detail and draws careful conclusions.

Minor detail: Abstract: Since it is mentioned in the abstract that renal function might improve it seems important to include also the non-significant p-value in the results of the abstract ( line 30).

Author Response

Thank you very much for your comments.

Minor detail: Abstract: Since it is mentioned in the abstract that renal function might improve it seems important to include also the non-significant p-value in the results of the abstract ( line 30). --> p-value was added in the abstract

Reviewer 2 Report

This is an interesting report describing using a German made Somatex vesicoamniotic shunting to treat LUTO and compared its efficacy with the US made Harrison set. The result revealed SOMATEX is superior to Harrison set.

In this part of the world, for example, Taiwan and Japan, we adopted Japan made double basket shunt, the advantages of these shunt over the more famous Harrison or Rodeck shunt are their finer caliber and it is less likely to dislodge or dislocate. The authors are advised to cite double basket catheter to this paper and also provide photographs of both shunts to enhance the readability of this work.

Author Response

Thank you very much for your comments.

In this part of the world, for example, Taiwan and Japan, we adopted Japan made double basket shunt, the advantages of these shunt over the more famous Harrison or Rodeck shunt are their finer caliber and it is less likely to dislodge or dislocate. The authors are advised to cite double basket catheter to this paper and also provide photographs of both shunts to enhance the readability of this work.

We added a paragraph citing the double basket shunt in the discussion and added a picture of the Harrison and Somatex shunts.